# Improved LiDAR Probabilistic Localization for Autonomous Vehicles Using GNSS

**DOI:** 10.3390/s20113145

**Published:** 2020-06-02

**Authors:** Miguel Ángel de Miguel, Fernando García, José María Armingol

**Affiliations:** Intelligent Systems Laboratory, Universidad Carlos III de Madrid, Av. de la Universidad, 30, 28911 Leganés, Madrid, Spain; fegarcia@ing.uc3m.es (F.G.); armingol@ing.uc3m.es (J.M.A.)

**Keywords:** localization, LiDAR, GNSS, Global Positioning System (GPS), monte carlo, particle filter, autonomous driving

## Abstract

This paper proposes a method that improves autonomous vehicles localization using a modification of probabilistic laser localization like Monte Carlo Localization (MCL) algorithm, enhancing the weights of the particles by adding Kalman filtered Global Navigation Satellite System (GNSS) information. GNSS data are used to improve localization accuracy in places with fewer map features and to prevent the kidnapped robot problems. Besides, laser information improves accuracy in places where the map has more features and GNSS higher covariance, allowing the approach to be used in specifically difficult scenarios for GNSS such as urban canyons. The algorithm is tested using KITTI odometry dataset proving that it improves localization compared with classic GNSS + Inertial Navigation System (INS) fusion and Adaptive Monte Carlo Localization (AMCL), it is also tested in the autonomous vehicle platform of the Intelligent Systems Lab (LSI), of the University Carlos III de of Madrid, providing qualitative results.

## 1. Introduction

Autonomous vehicle localization is the problem of estimating its position, determined by the *x* and *y* coordinates in a map and its orientation. This localization must be as accurate as possible since many vehicle’s modules, such as control or path planning strongly depend on how good it is. Errors in localization can cause the vehicle to have an undesirable behavior or to even not being able to follow the desired path. Localization techniques can be divided mainly in mapping or sensor based [1]. Global Navigation Satellite System (GNSS) information is commonly used to solve localization problems using a sensor. It provides a good global localization with no drift but it has to deal with some errors from different sources. Those errors can be generated due to the satellites themselves (e.g., clock inaccuracies or dilution of precision), interference in the satellite signal (e.g., signal jamming, satellite occlusion) or signal propagation errors. That last error source includes inaccuracies produced by different weather conditions in the ionosphere and troposphere earth layers, and by multipath interference, caused by the reflection of the satellite waves when the vehicle is surrounded by large obstacles (high buildings or trees) [2,3].

High precision GNSS receivers, like Real Time Kinematic (RTK) or differential GPS, improves accuracy considerably but they have commonly a very high cost and they don’t solve some accuracy errors. On the other hand, laser probabilistic localization methods, like Monte Carlo Localization (MCL) [4] can compare a precomputed map with the laser readings to acquire a precise position and orientation of the vehicle. The problem presented by this kind of algorithms is the opposite that with GNSS; in open environments with less map features, their accuracy decreases significantly. Here, particles of the filter would disperse generating different clusters of particles far away from the actual localization. This is known as the kidnapped robot problem and is a common localization error for probabilistic methods like MCL [5].

To solve those autonomous vehicle’s localization problems, many different solutions have been proposed. One of the most known solution includes the fusion of different localization sources using Kalman filter based methods, i.e., Extended Kalman Filter or Unscented Kalman Filter [6]. This method commonly fuses GNSS, IMU and other different odometry sources like the one generated by the wheels encoders or LiDAR/camera odometry.

Fusion filters based on Kalman fuses all the localization sources and generates a more accurate estimation of the position, but they depend strongly on the accuracy of the GNSS measurements, as it is the only absolute localization source i.e. the only one that does not have drift.

Furthermore, GNSS can suffer different problems, being one of the most common the multipath problem, which is a common reaserch topic where several works try to reduce it. In [7], a digital map of the environment is generated with OpenStreetMaps to prevent these errors and in [8], multipath is estimated and mitigated using a particle filter. However, the results don’t give enough accuracy and they depend on the information in OpenStreetMaps which is not always available.

Another solution adopted by some researchers consists in probabilistic localization, which can be performed using different sensors such as LiDAR [9], cameras [10,11] or magnetometers [12]. One of the most common method for this kind of solution is Monte Carlo Localization (MCL) [4]. MCL works as a probabilistic particle filter that uses the match between the LiDAR sensor and the map as a feature for each particle that determines the probability of existing in the next iteration. As an improvement for MCL, Adaptive Monte Carlo Localization (AMCL) outperforms classic (MCL) [13] as it uses Kullback-Leibler Distance (KLD) sampling to make the filter converge faster. Particle filter localization methods can be applied to autonomous vehicles, like in [9], that uses a 2D LiDAR and a map with the features of the road.

Both previously commented algorithms, GNSS and MCL have a good performance, but also have some drawbacks. To get the best from every algorithm, several works explore the idea of combining the two sources of information to improve the localization. Most of the works available in the literature, improve MCL by resetting the calculated position when it differs too much from the GNSS position, in other words, the kidnapped robot problem is corrected once it is detected. Those improvements consists of resetting methods, which makes MCL more robust. An example of this is [14], where the kidnapped robot is detected and then solved using the Expansion resetting method.

Different localization sensors are used to solve this problem like in [15], where GNSS is used to detect and solve it, or in [16], where WiFi signal detection provides information about the localization error. These methods give good results as they are able to detect and solve the kidnapped robot error in most of the cases. However, the time necessary to detect and solve this problem adds further error to the system, as during this time, the vehicle is driving with a wrong localization. That localization error is unacceptable for autonomous driving vehicles as it would result in wrong control or path planning commands and thus, incorrect behavior. Consequently, it seems a reasonable assumption that the prevention of the kidnapped problem, as it is intended in this work, would lead to better results.

The work of [17] gives a solution based on replacing particles with a low degree of laser fit on the map with new particles according to the probability density given by the sensor, but it can have localization problems in empty maps as no GNSS is used. In [18], GNSS data are used to generate new particles on the filter and to eliminate distant ones. However, this method doesn’t handle orientation, it only considers the position. Furthermore introducing particles based on GNSS data in all the cycles of the filter, would increase the noise of the localization and unfortunately no quantitative results of its performance are provided. As shown in [19], fusion of both sensors based on particle weight gives better results than adding new particles based on GNSS data. However, that work does not handle the kidnapped robot problem, as no strategies to recover are performed when the GNSS and particle filter positions differ considerably.

In contrast to all the approaches mentioned above, our method continuously uses GNSS data in the filter in both ways: modifying the weight of the particles, and injecting new ones if needed, avoiding kidnapped robot problem and making unnecessary to detect and reset states where the robot is badly localized. Furthermore, the method is designed to not replace directly the particles with new ones based on GNSS data, but to calculate its probability considering multiple parameters of both localization sources, making the particle filter more stable, an reducing noise generated by adding directly new particles on every cycle of the filter. Besides, we also consider orientation error when calculating the new probability, which is not done in most of the reviewed works.

The rest of the paper is structured as follows: Section 2 and Section 3 describe the software architecture and the method proposed respectively, Section 4 evaluates the localization with real data and in Section 5 the conclusions are exposed.

The proposed method is coded and tested with the KITTI Dataset [20]. The source code of this method is publicly available at https://github.com/midemig/gps_amcl so anyone can replicate the experiments described in this work.

## 2. Sensors and Software Architecture

This section describes the software architecture of the tests, including the configuration of the different modules used. This architecture includes the AMCL implementation for Robot Operating System (ROS) [21] from [4], the GNSS/INS based localization module [22] and the map generation module, in charge of the generation of the map, used by AMCL module.

### 2.1. Software Modules

In this section, the specific version of the software modules used in the comparison are described.

#### 2.1.1. AMCL

This algorithm outperforms original MCL [13] and is chosen to be the probabilistic LiDAR localization used. Specifically, it is used the AMCL ROS node implementation as it is a stable and maintained version of the algorithm. Most of the parameters’ configuration is set to the default values values, however, the most relevant ones are shown in Table 1, were the odometry model defines the equations that better describe the movement of the vehicle, the laser model describes the method used to calculate the probability of being at a certain position given a laser measurement and max particles and beams represents the maximum number of possible positions and laser beams around the vehicle respectively (more particles and beams can increase localization accuracy, but also computation time).

When the AMCL module is used in a map with a small number of reference obstacles surrounding the vehicle, the kidnapped robot localization error can easily appear, as shown in Figure 1, where multiple particles clusters appear due to the lack of features in the map, and the localization obtained by AMCL is inaccurate.

#### 2.1.2. Fusing Method

Kalman filter is commonly used for fusing localization data from different sources giving, as a result, a more accurate one. The software version used in this work is the robot localization ROS implementation [22]. Unscent Kalman filter is used as it is known to deal better with non-linearities in the filtering process [23]. In order to improve GNSS localization, it is fused with IMU data.

Robot localization package can fuse *n* different localization sources enhancing single-sensor based localization, and providing useful information such as the covariance of the position, which is of great relevance in this work.

#### 2.1.3. Map Generation

To use AMCL localization, a pre-generated map of the environment is generated using different Simultaneous Localization and Mapping algorithms [24]. As later on, GNSS information will be added to the localization step, this map must be generated with GNSS localization data. That means that a high precision GNSS receiver is needed, but only for the map generation. The Laser information is transformed accordingly to the GNSS position at each time step and then accumulated into the map using gmapping Simultaneous Localization and Mapping (SLAM) method [25]. The generated map (Figure 2) has all the features needed later by the AMCL algorithm to match a new laser scan with it.

### 2.2. Vehicle Sensor Setup

Although the results presented in the test section were performed using KITTI dataset, this algorithm can be implemented in any vehicle that fulfills some specified sensor requirements that are explained in this subsection:

#### 2.2.1. LiDAR

LiDAR Information is necessary to provide map information, although other 3D output may be used, the recommendation for this application, where accuracy is a key point, it is to use 3D Laser scanner technology. KITTI data provides a 64 layers 3D laser scanner. As 360 degrees 2D LiDAR scanner is needed to get environment information and match it with the pre-generated map using AMCL, the 3D laser scanner information is converted to 2D. This is done by choosing a minimum and a maximum height from the LiDAR, the multiple layers can be converted into a single 2D one. These parameters are selected to incorporate the maximum amount of features from the environment, but removing the ground plane, as it would only increase the noise in the particle filter. Other LiDAR configuration can be used, the test vehicle of LSI where this algorithm was implemented and tested is based on a 32 layers LiDAR, which provide similar results.

#### 2.2.2. GNSS Receiver

For this application, a Low-cost GNSS receiver can be used. Instead of high-cost RTK or differential GPS receivers, our method can work with lower accuracy in GNSS localization, making possible the generalization of these applications [26]. As the only localization information provided by KITTI dataset is the ground truth, it is considered to not have any zero error. Here, a low-accuracy GNSS receiver can be simulated by adding Gaussian noise to every ground truth measure, providing output similar to low-cost sensors [27]. The qualitative results provided on this test were performed using a low-accuracy GPS receiver, based on PixHawk technology.

#### 2.2.3. Inertial Measurement Unit (IMU) Sensor

IMU is the most common sensor fused with GNSS data, as it can provide position and orientation data. KITTI database provides IMU data with extrinsic calibration information, needed for the fusion. The PixHawk sensor unit used in the qualitative tests were performed using the PixHawk unit.

## 3. Method Description

Based on the original AMCL algorithm, several modifications are made to integrate GNSS data into the loop.

In this section, all those modifications are detailed and justified. Furthermore, the resulting algorithm is presented in Algorithm 1.

### 3.1. LiDAR Likelihood

The proposed solution computes a weight for each particle in AMCL by comparing the laser data transformed to each particle position with the map. In addition to that weight, our method computes a score of how accurate this particle is matched with the map. This score $s_{i}$ is computed using a Gaussian model [28,29] for the LiDAR data, following the expression:(1)$$s_{i}=\frac{1}{N}\cdot \sum_{n=1}^{N}\frac{1}{\sigma_{hit}\cdot \sqrt{2\cdot \pi }}\cdot e^{\frac{-z_{n}^{2}}{2*\sigma_{hit}^{2}}}\;$$
where *N* is the number of lasers of a laser scan, *z* is the distance from the laser hit point to the closest map occupied cell and $\sigma_{hit}$ is the standard deviation of the laser. With this expression, we can evaluate how good the LiDAR measurements match with the map in every particle position, and is later used to modify the probability of that particle to exist.

### 3.2. GNSS likelihood Estimation

In addition to weight calculated based on the matching of the sensor with the map, we add a second weight based on the GNSS Kalman filtered position estimation. As for the LiDAR likelihood, a Gaussian model is used to estimate the GNSS based weight of each particle $d_{i}$, but in this case, as the position received and each particle of the filter are three dimensional (*x*, *y* and ψ), the *n* dimensional Gaussian model is used,
(2)$$d_{i}=\frac{1}{{\left(2\cdot \pi \right)}^{\frac{3}{2}}\cdot {\left|\Sigma \right|}^{\frac{1}{2}}}\exp\left(-\frac{1}{2}{\left(x_{i}-\mu_{k}\right)}^{T}\Sigma_{k}^{-1}\left(x_{i}-\mu_{k}\right)\right)\;$$
where the received position is $\mu_{k}=\left(x_{k},y_{k},\psi_{k}\right)$ with covariance matrix $\Sigma_{k}$ and the position and orientation of each particle is defined with $x_{i}=\left(x_{i},y_{i},\psi_{i}\right)$. Using this model, orientation error is also considered when computing the GNSS based weight. Furthermore multiplying all the errors by the inverse of the covariance matrix in the exponential part, makes position and orientation errors scale-invariant. Here it is important to remark that the use of the covariance of the Kalman output allows reducing the importance of this weight when the quality of the information is low.

### 3.3. New Particle Weight Computed

To modify the weight of every particle so that Kalman filtered GNSS data are incorporated into the filter, the new weight of every particle is calculated by making use of the weights computed in (1), (2), and the following equation:(3)$$w_{i-new}=w_{i}\cdot s_{i}\cdot k_{l}+d_{i}\;$$
where $w_{i}$, the weight calculated by original AMCL, is modified in order to incorporate GNSS probabilistic data. The weight $k_{l}$ is a constant and it is added to balance the importance of each source of information. It is empirically set to 200, where it gives the best results in all the environments tested. After this weight modification, they are normalized to make the sum of all the weights equal 1.

### 3.4. New Particles Generation

As the particle filter eliminates particles accordingly to its probability, the original AMCL method adds new particles randomly distributed in the map, based on two parameters that define how often it is needed to add those particles. A different function was defined to generate new particles $X_{new}$ near GNSS Kalman position and to determine the probability of generating those new particles. The following expression defines how the new particles are generated to follow a normal distribution centered in $\mu_{k}$ with covariance $\Sigma_{k}$.
(4)$${\left(\begin{array}{ccc} x_{new} & y_{new} & \psi_{new} \end{array}\right)}^{T}=\lambda_{k}^{\frac{1}{2}}\phi_{k}\cdot R+\mu_{k}\;$$
where *R* is a random vector with distribution N[0,1] and λ, ϕ are the diagonal matrix of eigenvalues and the eigenvectors matrix of the covariance $\Sigma_{k}$ respectively. Besides, the probability *p* of generating a new GNSS based particle at each cycle is determine as:(5)$$p\left(X_{new}|x,\mu \right)=\left\{\begin{array}{cc} d_{mean}, & \mathrm{if}\;d_{mean}>0\\ 0, & \mathrm{otherwise} \end{array}\right.\;$$
with
(6)$$d_{mean}=p_{max}-\frac{1}{N}\cdot \sum_{n=1}^{N}d_{i}\;$$
where $p_{max}$ is the maximum probability allowed to generate new particles at each filter cycle and is experimentally set to 0.01.

The addition of particles does not increase the noise in the final localization given by the particle filter, as they are only added when the filter particles begin to considerably differ from the GNSS localization, and the newly generated number never goes beyond the limit of 1%.This situation avoids the creation of GNSS based particles when the filter provides accurate detection, these are generally particularly difficult situations for the GNSS such as urban canyons.
**Algorithm 1:** New weight and resample of filter particles
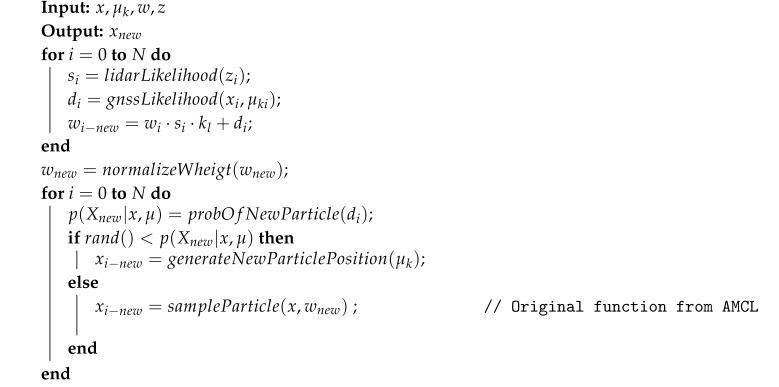


## 4. Experimental Results and Discussion

Two different group of experiments were performed. On the one hand, the proposed method is tested using the KITTI odometry dataset for quantitative results. On the other hand, the LSI platform for autonomous driving is used for qualitative results [30].

### 4.1. Dataset

The KITTI dataset is commonly used to test different odometry algorithms, and compare them as it includes calibrated data from different sensors (cameras, LiDAR and IMU) and precise ground truth for localization. The tests performed using this dataset compare the ground truth localization accuracy with the three following methods:Kalman filtered GNSS and IMU. As covariance of GNSS localization is set fixed, the mean error value is displayed for visualization purpose.AMCL original implementation.Proposed method, having the same odometry source as original AMCL and the same parameters configuration.

For every position and orientation, euclidean distance and orientation error are compared with the closest one in time of the ground truth (interpolating if necessary). KITTI dataset gives multiple sequences in different scenarios. For this evaluation residential sequences were selected, since they combine narrow and empty streets, including both types of scenarios. For every sequence, first, a map is generated using the localization ground truth and the LiDAR data described in Section 2.1.3. The experiments are designed to compare the proposed method with the other two considering the worst-case scenario for our method where, at least, it needs to give similar results to the comparing methods. Then a normal situation is tested to quantify the improvements of the proposed method.

Considering this, the following three scenarios are tested.

### 4.2. Empty Map

This scenario refers to situations with a low number of obstacles, i.e. lack of references for the LiDAR points to match. This is considered to be the worst scenario for AMCL. However, GNSS has better accuracy when it is not surrounded by obstacles that interfere with observations. Localization error is compared for Kalman filtered GNSS and the proposed method. As it is shown in Figure 3, localization error provided by our approach is very similar to the GNSS based, giving almost the same values for position and orientation. This is according to the expected as the proposed method can not improve localization accuracy with no laser information, but it proves that in empty environments it would perform as good as GNSS based localization.

### 4.3. GNSS Challenging Scenarios

This scenario describes the opposite problem. In an environment full of objects or an urban environment with urban canyons, GNSS localization would fail or would give noisy measurements with high covariance. In these cases, AMCL algorithm gives better results thanks to a map with numerous features and objects to match with the laser points. Localization is compared in this scenario for AMCL and the proposed method. As shown in Table 2, localization errors are similar. When no GNSS localization is received, or it has a high covariance, the proposed method, as it is designed, has a very similar behavior as original AMCL.

### 4.4. Mixed Environments

The last scenario tested is a more common one where GNSS data are received with an acceptable covariance, and the map has features unequally distributed on it, generating places with more objects and empty places. The three algorithms are tested and compared in this scenario using the KITTI dataset residential sequences, that include more than 45 min of recorded data in a residential environment, but only the most relevant ones are discussed in this section.

#### 4.4.1. AMCL

When evaluating AMCL performance two different localization issues can be identified. The first one, as shown in Figure 4, presents an increase in the error due to multiple particles’ clusters that make the filter jump from one to another, increasing the error but finally converging again to the true localization. The second problem can be seen in Figure 5, where the kidnapped robot problem appears around the second 180, where the localization jumps to a similar place in the map.

#### 4.4.2. GNSS

It is set to have a constant covariance error, so this localization maintains the same localization error along all the sequence. In real environments, Kalman results could be even worse as GNSS position error could be even higher in narrow streets.

#### 4.4.3. Proposed Method

The proposed method, gives a better performance along all the path, improving localization mean error compared to AMCL and GNSS. It outperforms the other algorithms in terms of accuracy and stability as it has a localization error better or equal to Kalman localization method, and does not present kidnapped robot localization problem.

Figure 4 and Figure 5 present the localization error along the sequence and the mean error values for sequences 36 and 34 respectively, showing how the proposed method outperforms original AMCL avoiding the kidnapped robot problem, while maintaining the localization error below the GNSS one.

Moreover, Table 3 compares the error of the proposed method as the accuracy of the GNSS localization decreases. As it is shown, for high GNSS accuracy, the error is close, but as the GNSS error increases, our method is able to reduce it using the laser information, giving always better results than the original AMCL algorithm.

### 4.5. Real Platform Qualitative Results

In addition to the evaluation using KITTI database, the proposed method is also tested in our research platform as shown in Figure 6. As this platform does not include ground truth localization information, only a qualitative analysis of the performance of the proposed method is possible giving, as a result, a stable localization accurate enough to control the vehicle, solving the kidnapped robot that this platform showed before the implementation of this method when using AMCL, and improving the localization accuracy of the GNSS receiver.

## 5. Conclusions

In this work, a novel localization method for autonomous vehicles is proposed. It consists of a modified version of AMCL where GNSS data are integrated. As it is shown in Section 4, this method combines the strengths of the two combined localization methods, without compromising accuracy at any scenario, urban and non-urban. The results prove a good performance and a smooth transition from using GNSS data when the map is featureless to using LiDAR and map data when GNSS localization is not accurate, improving localization when both sources are available. The improvements depend on the environment but the proposed method always gives better results, or, in the worst case, the same results. It also gives a low-cost solution for different enviroments using low-cost sensors such as one layer LiDAR or low accuracy GNSS receiver compared to systems that use High-cost RTK or differential GNSS receivers and multiple layer LiDARs, making possible the generalization of these applications. The method is tested and evaluated with a well-known database (KITTI) which is usually used to evaluate autonomous vehicles perception and localization algorithms and with a real platform, improving its performance thanks to a better localization. Besides, the source code of the method is published so it can be tested and improved by anyone. 

## Figures and Tables

**Figure 1 sensors-20-03145-f001:**
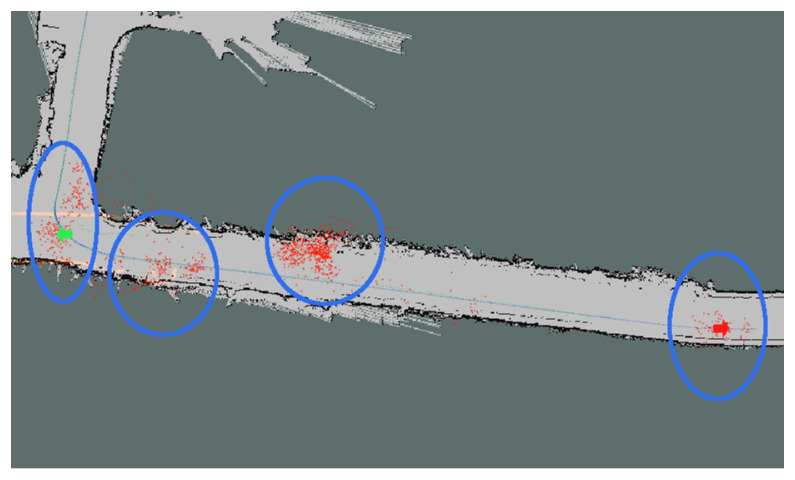
AMCL kidnapped robot localization error. Different clusters rounded in blue, red dots are the filter’s particles and red and green arrows are ground truth.

**Figure 2 sensors-20-03145-f002:**
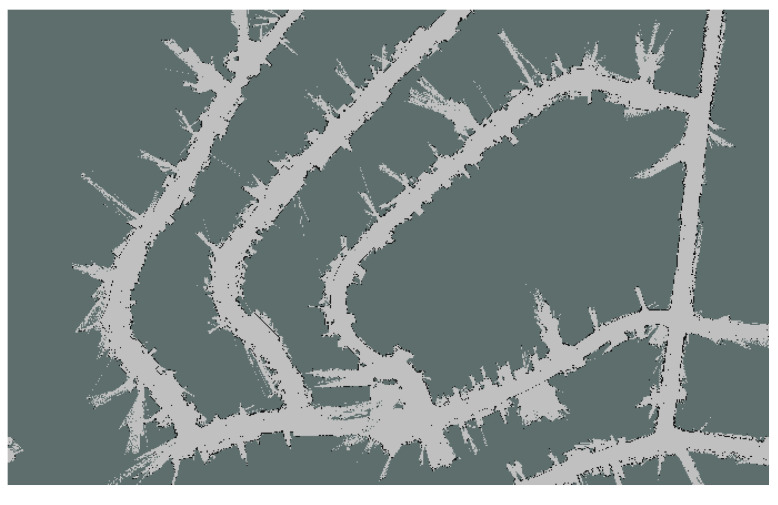
Map generated of one of the KITTI sequences.

**Figure 3 sensors-20-03145-f003:**
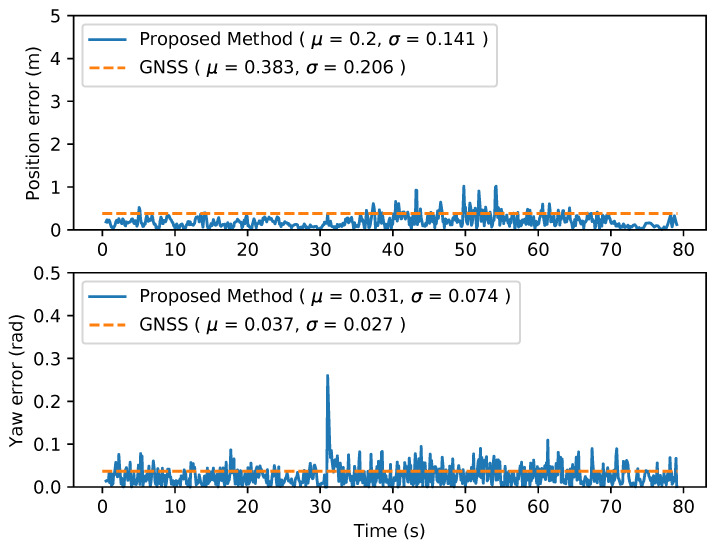
Comparison between proposed method with no LiDAR information but Global Navigation Satellite System (GNSS) localization. (GNSS is plotted as constant of mean value for visualitation purpose.)

**Figure 4 sensors-20-03145-f004:**
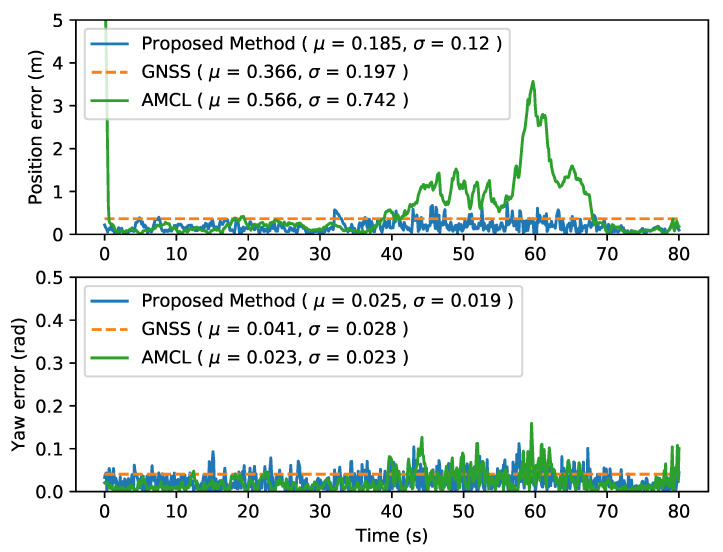
Localization error of the three methods described in KITTI sequence 36. (GNSS is plotted as constant of mean value for visualitation purpose).

**Figure 5 sensors-20-03145-f005:**
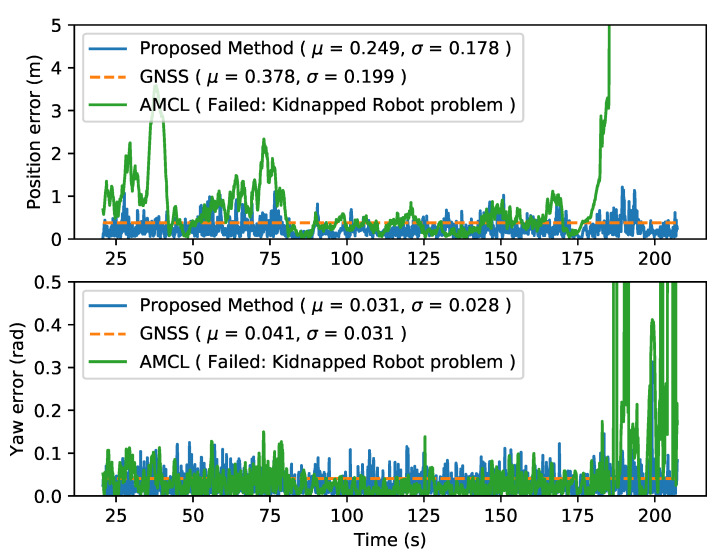
Localization error of the three methods described in KITTI sequence 34. (GNSS is plotted as constant of mean value for visualitation purpose).

**Figure 6 sensors-20-03145-f006:**
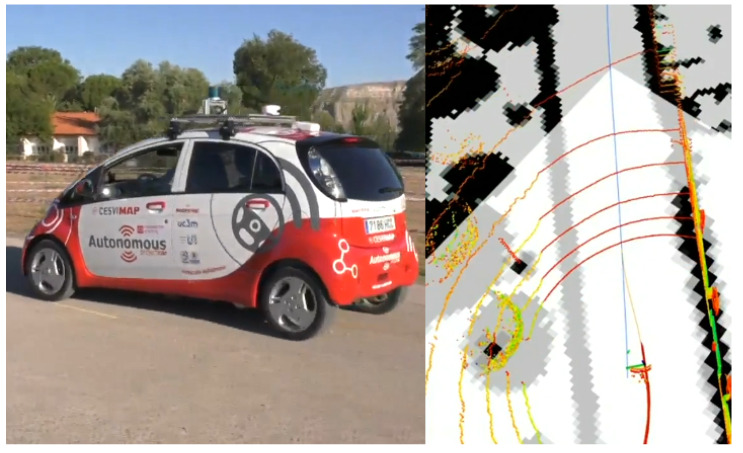
Our autonomous vehicle platform testing the proposed method (left), and a visualization of the LiDAR pointcloud and the map (right).

**Table 1 sensors-20-03145-t001:** Adaptive Monte Carlo Localization (AMCL)’s most relevant parameters.

Parameter	Value
Odometry model	differential
Laser model	likelihood filed
Max particles	2000
Max beams	360

**Table 2 sensors-20-03145-t002:** Comparison between proposed method and AMCL with no GNSS information.

Method	Position Mean (m)	Position Std (m)	Yaw Mean	Yaw Std
Proposed	0.513	0.856	0.033	0.025
AMCL	0.566	0.742	0.023	0.023

**Table 3 sensors-20-03145-t003:** Evaluation of the proposed method errors for different GNSS mean errors in sequence 34.

GNSS Mean (m)	Position Mean (m)	Position Std (m)	Yaw Mean	Yaw Std
0.127	0.141	0.137	0.015	0.016
0.374	0.186	0.11	0.028	0.024
1.256	0.367	1.767	0.029	0.129
6.256	0.496	1.963	0.023	0.116
12.600	0.554	2.115	0.026	0.151
37.581	0.593	2.495	0.032	0.196
AMCL	Robot kidnapped error			

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
