# Peer review of "Improved LiDAR Probabilistic Localization for Autonomous Vehicles Using GNSS"

_sensors, 2020, doi:10.3390/s20113145_

Round 1
Reviewer 1 Report
See the attached file.

Author Response
General comments and discussion questions
"Is there really only a multipath problem in GNSS? The authors focus their attention only on this problem. Multipath in the case of phase measurements gives a small error, on the order of 1⁄4 of the wavelength (about 5 cm), see e.g. Global Positioning System (GPS): Theory and Practice by Hofmann-Wellenhof et al., 1994). On the other hand, in the case of difficult terrain with partly shadowed sky, too few visible satellites and too high DOP coefficients (e.g. PDOP) can cause bigger problems than multipath.”
Authors’ response: We would like to thank the advice of the reviewer, which helped us to enhance the quality of the paper with the comments provided. Regarding to the aspect pointed by the reviewer in this comment, although the original paper discussed briefly different GNSS accuracy problems, this explanation may not have been extensive enough and parts of the text focus too much in this particular problem. Thus, the reference to multipath has been changed to accuracy errors to be more generic.
“b. Why is it so important to use "low-cost" GNSS receivers? Usually there is a lot of expensive equipment in vehicles for autonomous navigation, why is the cost of a GNSS receiver so important?”
Authors’ response: The reviewer make an interesting comment of the usability and interest of the approach proposed. We believe that the reduction of the cost of the sensors is important to generalize the use of these applications, and to make possible the industrial production of these technologies.The proposed method can work with lower accuracy sensors, which often have a lower cost. In order to clarify this aspect, further explanation was added to the text to justify the use of low-cost sensors.
“c. Is 80 sec of measurements (as in the case of data shown in Figures 3 and 4) or 200 sec (Fig. 5) a sufficiently long observation period to reliably draw conclusions about the correct operation of the method? On the charts, to ensure their readability, you can leave such short periods, but in my opinion, it would be worth at least to discuss the results of longer observation periods and their results.”
Authors’ response: We appreciate the opportunity to clarify this point. The method was tested in the kitti dataset as this is the reference benchmark for many of the work published in the field. However there is only a set of sequences that combine both interurban and urban scenarios, i.e. scenarios for which the approach was specifically designed, the tested sequences included in the work were reduced to these specific scenarios, further tests were included to prove the efficacy of the work in real life scenarios in the other results sections. A clarification is included in the text to note that the method was tested in different sequences.
“d. Section 5.5 "Real platform qualitative results" basically adds nothing to the discussion. It should be either omitted or significantly supplemented. If the results obtained allow the authors to conclude that the method ensure the "stable localization precise enough to control the vehicle", then it is worth describing on what basis such a conclusion was drawn. In addition, the right side of Figure 6 needs clarification.”
Authors’ response: We agree with the reviewer that this section does not provide proof of the performance of the approach proposed, however we are afraid that we do not share with the reviewer the opinion of the importance of this section. We find it of vital importance to prove that the approach is currently working in the test platform of the LSI. Hence, the purpose of this subsection is to prove that the method is applicable to a real vehicle, and not only to the simulated or recorded data. Due to the lack of a ground truth, no quantitative data of the accuracy of the method can be provided on this section, but instead, we can prove that the performance of the vehicle with this method is improved in relation to the previously used methods. A clarification and further description of the figure are included in this version which we expect helps to identify its importance.
2. Detailed remarks
"Lack of explanation of some abbreviations (like for example PF in line 87, ROS in l. 99 etc.)."
"Kalman is often written with a lowercase letter, but we must remember that it is a surname (eg. lines 3, 116, etc.)."
Authors’ response: Thanks to the help of the reviewer, the mentioned mistakes and typos were corrected in this new version.
“c. In table 1 - there is enough space to add column titles, and call the models applied without using abbreviations, and even briefly describe them, because they are the "most relevant".”
Authors’ response: Following the recommendations of the reviewer, in this new version the configuration parameters are described. Furthermore, all tables abbreviations have been extended for better comprehension.
“d. Line 147 – does adding Gaussian noise correctly reflect reality? Usually, in GNSS
measurements the problems are caused by not eliminated systematic influences (like tropospheric delays, ionospheric delays, multipath), not random.”
Authors’ response: Although we agree with the reviewer that GNSS problems are mostly related to not eliminated systematic influences. We followed the reference provided that indicates "If the receiver is sufficiently narrow band, the noise at the receiver output can be reasonably assumed to be modeled well as a Gaussian process". Hence we assumed that a more real model for GNNS errors would not change the results provided. We added text in the document to clarify this point.
“e. Please complete the explanation of the weight update process in 4.3. The equation (3) for new weights uses s i from formula (1), d i from formula (2), k l is a constant with empirically given value and there also occurs the old weight w i . It is not explained how the old weight was obtained (or how this weight update process was initiated).”
Authors’ response: Thanks to the help of the reviewer, the mentioned error has been corrected. w_i refers to the weight that AMCL algorithm with no modifications would have provided. In this new version the text was modified to clarify this point.
“f. In (1) σ hit is not the covariance, as it was said. It is square root of the variance, it can be also called standard deviation.”
Authors’ response: Following the reviewers recommendation, this issue have been corrected in the text.
“g. The reader will be grateful for a few words to explain the formula (4) and justify its form. In addition, matrices φ,λ should also have indexes k?”
Authors’ response: Following the recommendation of the reviewer, this part of the text has been corrected to allow full understanding of the solution. The formula explains how to get random samples that follow normal distribution with covariance sigma. The following changes were applied to this part of the text:
- K indexes have been added.
- A few words to clarify the expression have been added.
“h. It seems that the structure of the subdivision in chapter 5 is incorrect. Section 5.1 is called “Quantitative Results”, and it is followed by subsections 5.2, 5.3, 5.4, which also present quantitative results. I suggest changing this structure to a more logical one.”
Authors’ response: Following the recommendation of the reviewer section 5 ( now section 4) have been redesigned, to avoid this misinterpretation now section 4.1 title has been changed to “dataset”, as it describes the dataset used.

Reviewer 2 Report
A few corrections in the text. Please note that I am not a native English speaker, so take these corrections with due precaution !
Line 10 : quantitative -> qualitative
Line 20: I would add satellite occlusion to the list of interference in the signal
Line 50: "being one of the most common the multipath problem" -> "one of the most common being the multipath problem"
Line 92: "to not to replace" -> "to not replace"
Line 118: "fusion" -> "fuse"
Line 193: It lacks a final period.
Line 256: "comparing" -> "other"
Line 275: "at" -> "in"
Line 279: "gives"-> "give" (maybe you should reword the sentence)
Line 279: "worse" -> "worst"
Line 282: "with well-known" -> "with a well-known"
Line 284: "due to" -> "thanks to"
Author Response
“A few corrections in the text. Please note that I am not a native English speaker, so take these corrections with due precaution !”
Authors’ response: We appreciate the comments of the reviewer. The corrections given by the reviewer have been carefully followed and the text was adapted accordingly. Changes to the first version have been highlighted in the text to facilitate their tracking by the reviewers.

Reviewer 3 Report
The manuscript presents a method of autonomous vehicle localization using a modification of the Monte Carlo localization algorithm by adding GNSS data. The algorithm is tested using both KITTI odometry dataset and data acquired in the vehicle platform of the Intelligent Systems Lab (LSI), of the University Carlos III of Madrid.
As autonomous vehicle localization has become popular, the topic discussed is relevant and the method proposed by the authors is innovative as well.
Nevertheless, in my opinion this document needs to be improved before it can be accepted for publication.
General comments
The paper's structure is not standard and needs to be revised.
The paper begins with a short introduction followed by a section 2 titled "related work". In both sections the state of the art is described (in little depth) and both end with a description of the authors' work. In my opinion, the Introduction should be a single section (related work should be part of it) and completely rewritten, describing the state of the art, the drawbacks, the challenges and the innovation of the authors' work compared to the others.
The "architecture" section should also be unified with the section "method description" and revised.
There are too many sub-section. A discussion section is completely missing.
I recommend extensive editing of English language and style. in the attached pdf file I have highlighted in yellow some words/sentences that in my opinion sound wrong.
A few detailed comments are in the attached pdf file.

Author Response
“The paper begins with a short introduction followed by a section 2 titled "related work". In both sections the state of the art is described (in little depth) and both end with a description of the authors' work. In my opinion, the Introduction should be a single section (related work should be part of it) and completely rewritten, describing the state of the art, the drawbacks, the challenges and the innovation of the authors' work compared to the others.”
Authors’ response: First we would like to thank all the efforts made by the reviewer to provide feedback of our work. The different aspects commented allowed us to improve the quality of the paper. We expect that this new version fulfills the expectations. Regarding to this issue, introduction and related work sections have been merged into one unique section. Furthermore, some changes have been made to the text to improve coherence and cohesion of both sections.
“The "architecture" section should also be unified with the section "method description" and revised. There are too many sub-section. A discussion section is completely missing.”
Authors’ response: We have tried to follow as far as possible the recommendation of the reviewer regarding to this point. The discussion section is unified with the results section under the name “Experimental results and discussion”. The results are discussed one by one as they are presented in the text.
“I recommend extensive editing of English language and style. in the attached pdf file I have highlighted in yellow some words/sentences that in my opinion sound wrong.
- A few detailed comments are in the attached pdf file:”
Authors’ response: Following recommendation of the reviewer, we have rewritten the highlighted sentences, and follow the comments of the attached pdf (move figures, references, etc). The changes in the new version are highlighted in yellow to allow the reviewer to identify them.
“"Architecture" is not a appropriate section's title”
Authors’ response: Following reviewer advice, section name was changed to "Software Architecture"
“In a methodological section it is not appropriate to recall what is written in Introduction. The section should contain a description of the methods used in the paper.”
Authors’ response: We followed the recommendation of the reviewer, hence the methods are described or referenced to the original paper in the section, without references to the Introduction.

Round 2
Reviewer 3 Report
Thank you for giving me the opportunity to reevaluate the revised manuscript. The comments were thoroughly answered, and the improvement of the manuscript is marked. I recommend publication in this form.